# Nutritional deficiency in scarlet eggplant limits its growth by modifying the absorption and use efficiency of macronutrients

**Gelza Carliane Marques Teixeira**[1]*, **Renato de Mello Prado**[1], **Kamilla Silva Oliveira**[1], **Antonio Carlos Buchelt**[2], **Antonio Márcio Souza Rocha**[3], **Michelle de Souza Santos**[1]

1 Department of Agricultural Sciences, São Paulo State University (UNESP), Jaboticabal, São Paulo, Brazil,
2 Department of Agronomy, Mato Grosso State University (UNEMAT), Alta Floresta, Mato Grosso, Brazil,
3 Department of Technology, São Paulo State University (UNESP), Jaboticabal, São Paulo, Brazil

* gelzacarliane@hotmail.com

**Data Availability Statement:** All relevant data are within the manuscript and S1 Data.

## Abstract

The intensity damages caused by nutritional deficiency in growing plants can vary with nutrients. The effects caused by nutrient omission in the plant nutritional efficiency in relation to the absorption and use of the missing nutrient, and the reasons why these damages reflect in other nutrients have not yet been reported in the culture of scarlet eggplant. A better understanding of the nutritional mechanisms involved may clarify why certain nutrients cause greater limitations than other during plants growth. Thus, this study was designed with the aim of evaluating the damages caused by macronutrients deficiency in the culture of scarlet eggplant in the accumulation of these nutrients, nutritional deficiency, plants growth and in visual symptoms. The experiment was carried out in a controlled environment where plants were cultivated in a hydroponic system. Treatments consisted of supplying a complete Hoagland and Arnon solution (CS), and other nutrient solutions with individual omissions of nitrogen (-N), phosphorus (-P), potassium (-K), calcium (-Ca), magnesium (-Mg) and sulphur (-S). When a nutrient deficiency arose, nutritional analyses, growth and visual symptoms were analyzed. The omissions of N, S and K in the nutrient solution resulted in lower accumulation of all macronutrients in both the above and below ground biomass. Individual omissions resulted in nutritional imbalances with reflexes in the absorption efficiencies and use of the missing nutrient, as well as of other nutrients, revealing that the metabolism involves multiple nutritional interactions. Losses of nutritional efficiencies of macronutrients caused detrimental effects on plants growth, with reduced height, stem diameter, number of leaves, leaf area, and biomass production in above ground and below ground. From the losses in production in above ground biomass, the order of macronutrients limitation was N, S, K, Ca, Mg, and P, with reductions of 99, 96, 94, 76, 51 and 46%, respectively, in comparison to plants cultivated in CS. The most limiting nutrients were N, S, and K, seen that its deficiencies affected the metabolism of all other nutrients. This study demonstrates the importance of an adequate nutritional management of N, S, and K in the cultivation of scarlet eggplant.

**Funding:** This study was financed in part by the Coordenação de Aperfeiçoamento de Pessoal de Nível Superior - Brasil (CAPES) (Authors benefited GCMT and RMP). The funders had no role in study design, data collection and analysis, decision to publish, or preparation of the manuscript.

**Competing interests:** The authors have declared that no competing interests exist.

## 1. Introduction

The scarlet eggplant (*Solanum gilo*) belongs to the family Solanaceae and its cultivation is intended for human consumption, due to its nutritional value [1]. This species is commonly recommended in healthy diets, as it has antioxidant properties and low caloric value, being considered a source of calcium, phosphorus, iron, vitamins C and B5, flavonoids, alkaloids, and steroids [2]. The elevated nutritional value stimulates the consumption of this vegetable of high economic importance, which has a planted area of 839 hectares where 21865 tonnes are yearly produced in the State of São Paulo, generating an approximate income of U$4,800 [3].

The proper nutrition of eggplant plants improves its visual and nutritional quality, as well as its flavor [4]; however, plant nutrition is a complex factor, due to the interactions between nutrients. These interactions can affect the processes of nutrient absorption and use and thus reflect in the mineral composition of crops, and consequently in its nutritional status, final production and quality [5]. Such deficiencies can cause nutritional imbalances, leading to the occurrence of deformities in organic tissues, reduced leaf area, limited growth and dry matter production [6–8]. Therefore, it is necessary to investigate the mechanisms involved in scarlet eggplant nutrition, as well as the symptomatology caused by the deficiency of macronutrients in this species with high agronomic potential.

The symptoms and severity of nutritional deficiency in plants may vary due to several biological functions and interactions that occur between nutrients and the environment [9], thus reflecting in its growth. For instance, a plant that is deficient in K inhibits the biological functions of this nutrient in the plant, such as the activity of multiple enzymes, some of which are involved in the proteins synthesis [8, 10]. This occurrence may hamper the metabolism of other nutrients, such as N, which plays an important role in relation to the increment of protein synthesis, but this function is in turn interrupted due to the low activity of enzymes involved in the assimilation of N [5, 8]. Therefore, even though the plant has an adequate supply of N via the nutrient solution, its metabolism could be impaired because of a lack of K, and the use efficiency of N will be low, increasing the biological damage in the organism [4]. These biological interactions occur with other nutrients that affect its nutritional efficiencies, and consequently the severity of deficiency symptoms that vary according to the species.

The effects of macronutrients deficiency in scarlet eggplants were poorly investigated to date, with one limited study developed by Haag et al. [11], in which the authors did not investigate nutritional interactions. Studies involving the nutritional deficiency of most species focus only in the evaluated nutrient and do not consider its interaction with other nutrients [7]. Thus, it is important to advance research in order to elucidate such interactions in plants, optimize fertilization and improve the efficiency of nutrients use by crops [12].

The hypothesis of this study is that the extent of biological damages caused by nutritional deficiency in scarlet eggplant plants depends on the omitted nutrient in the nutrient solution, associated with its interaction with other nutrients, as it may affect the efficiency of nutrients absorption and use by the plant. In case this hypothesis is accepted, it would be possible to comprehend the reasons why certain nutrients cause greater limitations in the growth of this species.

Thus, the aims of this study were to evaluate the effects of macronutrients omissions in the growth of scarlet eggplants and in the nutritional mechanisms of this culture, as well as describing the visual symptoms presented by plants under macronutrient omission.

## 2. Material and methods

The experiment was carried out in a greenhouse at the São Paulo State University (UNESP)—Campus of Jaboticabal, Brazil, from August to October 2018. Temperature and humidity data

were collected throughout the experimental period. There were high variations in the minimum (25 ± 5%) and maximum relative humidity (80 ± 5%), as well as minimum (18 ± 2°C) and (35 ± 5°C) temperature throughout the experimental period.

## 2.1 Treatments and experimental design

The treatments consisted in providing the Hoagland and Arnon [13] solution with all nutrients, also called complete solution (CS) (containing N, P, K, Ca, Mg, S, B, Mn, Zn, Cu, Fe and Mo), and the solution with selective nutrients being omitted (-): nitrogen (-N), phosphorus (-P), potassium (-K), calcium (-Ca), magnesium (-Mg) and sulfur (-S). The experimental design was entirely randomized with four replicates per treatment.

Scarlet eggplant seeds were sown in polystyrene trays containing vermiculite. After emerging, seedlings were irrigated daily with distilled water, and 20 days after emergence, these were transplanted into 0.7 dm$^3$ plastic pots filled with washed sand. The pots had holes in their bottoms and they were placed inside a plastic tray containing the nutrient solution, which could reach the roots by capillarity.

The source of iron (Fe) of the nutrient solution that was used was chelated Fe (Fe-EDDHMA). The concentration of nutrients in the solution was maintained at 15% throughout the first week of cultivation, being raised by 15% on a weekly basis, until it reached a concentration of 60% in the fourth week, as it remained until the end of the experiment. Its pH was maintained between 5.3–5.7.

After transplantation, the plants were cultivated with the complete nutrient solution during 14 days. After this adaptation period, nutrient omissions were imposed and the plants were then cultivated in this condition until the occurrence of deficiency symptoms that are characteristic of each nutrient. This occurred seven days after the start of omission (DAO) for plants grown in -N (21 days after transplanting—DAT); at 21 DAO for plants grown in -S and -K (35 DAT); at 30 DAO for plants grown in -Ca and -Mg (44 DAT); and at 33 DAO for plants grown in -P (47 DAT).

## 2.2 Analysis performed

Plants' height was evaluated using a graduated scale, considering from the base to the top of the plant. Stem diameter was determined with the aid of a digital caliper at 2 cm from the base of the plant, while the number of leaves was obtained by counting the completely expanded leaves.

Leaf area was measured with the aid of an equipment L-3100, LICOR, USA. At the end of the experiment, plants were separated into aerial parts (above ground biomass) and roots (below ground biomass). All plant materials were washed with tap water, submerged into a neutral detergent solution (0.1% v/v), then in a hydrochloric acid solution (0.3% v/v) and washed again in deionized water. Subsequently, these samples were dried in an oven with forced air circulation (65 ± 5°C) until a constant weight was obtained.

All samples of plant material were ground before analysis. The N content was determined by adding concentrated sulfuric acid to samples, followed by distillation and titration with sulfuric acid [14]. The levels of P, K, Ca, Mg, and S were determined by the digestion of samples, using a digestive mixture of perchloric and nitric acid (1:2), with readings of K, Ca, and Mg performed in spectrophotometry of atomic absorption with air-acetylene flame, while P and S readings were carried out by means of spectrophotometry [14]. The accumulation of each nutrient was calculated based on the plant biomass.

In addition, the absorption and use efficiency of nutrients were calculated using the data of macronutrients accumulation and plant biomass, as recommended by Fageria and Baligar

[15]. For this purpose, distinct equations were used to calculate the absorption efficiency: (accumulation in the whole plant/biomass of root); and use efficiency: ((biomass of whole plant)$^2$/accumulation in the whole plant). Regarding the visual effects on plants, these were daily monitored for symptoms of nutritional deficiency, and representative images of the symptoms were acquired.

### 2.3 Statistical analysis

All data were submitted to a variance analysis by the F test, and means were compared by the Tukey's test at a 5% probability level. Statistical analyses were carried out with the aid of the software SAS® (Cary, NC, USA). The data were also subjected to hierarchical cluster analysis as described by Teixeira et al. [16]. For this purpose, data were standardized by the following equation: $Zij = (Xij-Xj)/Sj$, wherein j = number of variables; i = number of treatments; Zij = standardized value of Xij; Xj and Sj = mean and standard deviation of the variables, respectively. Euclidean distance was used as a similarity coefficient and the UPGMA method (unweighted pair-group method using arithmetic averages) as a group connection algorithm. The statistical tests were performed using the free software environment R and the package "pheatmap".

## 3. Results

The omission of macronutrients in the nutrient solution reduced the accumulation of N, P, K, Ca, Mg and S in both above (Fig 1) and below ground biomass (Fig 2), highlighting the precision of this study in using a decontaminated nutrient solution. Thus, the accumulation of macronutrients in the aerial parts of plants (mg per above ground biomass) under CS in relation to each omission (-) was 581/4 for $CS_N$/-N, 96/15 for $CS_P$/-P, 896/6 for $CS_K$/-K, 192/7 for $CS_{Ca}$/-Ca, 77/8 for $CS_{Mg}$/-Mg and 58/1 for $CS_S$/-S, respectively (Fig 1).

The omission of each macronutrient affected the nutritional balance and the growth variables of plants in specific ways. The omission of N reduced the accumulation of N in the above (Fig 1A) and below ground biomass (Fig 2A). Additionally, plants cultivated in the condition -N presented a lower accumulation of P (Figs 1B and 2B), K (Figs 1C and 2C), Ca (Figs 1D and 2D), Mg (Figs 1E and 2E), and S (Figs 1F and 2F) in both above and below ground biomass, in comparison to the plants cultivated under all other conditions.

Both the absorption and use efficiencies of N were lower in plants under N deficiency in comparison to CS (Figs 3A and 4A). The absorption and use efficiencies of P (Figs 3B and 4B), K (Figs 3C and 4C), Ca (Figs 3D and 4D), Mg (Figs 3E and 4E), and S (Figs 3F and 4F) were also reduced in plants cultivated in the condition of–N, in comparison to CS.

These effects resulted in impaired growth, reducing its height (Fig 5A), stem diameter (Fig 5B), number of leaves (Fig 5C), and leaf area (Fig 5D), as well as the accumulation of above (Fig 5E) and below ground biomass (Fig 5F).

Nitrogen was the most limiting nutrient for the growth of scarlet eggplant plants, with losses observed in the dry matter production in both the above and below ground biomass, from up to 99 and 97% in relation to CS, respectively (Fig 5E and 5F). The visual symptoms of N deficiency were characterized by smaller height and generalized chlorosis of leaves from the lower third, which subsequently evolved to all other leaves (Figs 6A and 7A).

The omission of P also reduced the accumulation of this nutrient in the above (Fig 1B) and below ground biomass (Fig 2B), being accompanied by lower accumulations of N (Figs 1A and 2A), K (Figs 1C and 2C), Mg (Figs 1E and 2E) and S (Figs 1F and 2F) in both above and below ground biomass, and Ca in below ground biomass (Fig 2D), in relation to CS. Plants that were deficient in P presented the lowest absorption efficiency of this macronutrient in comparison

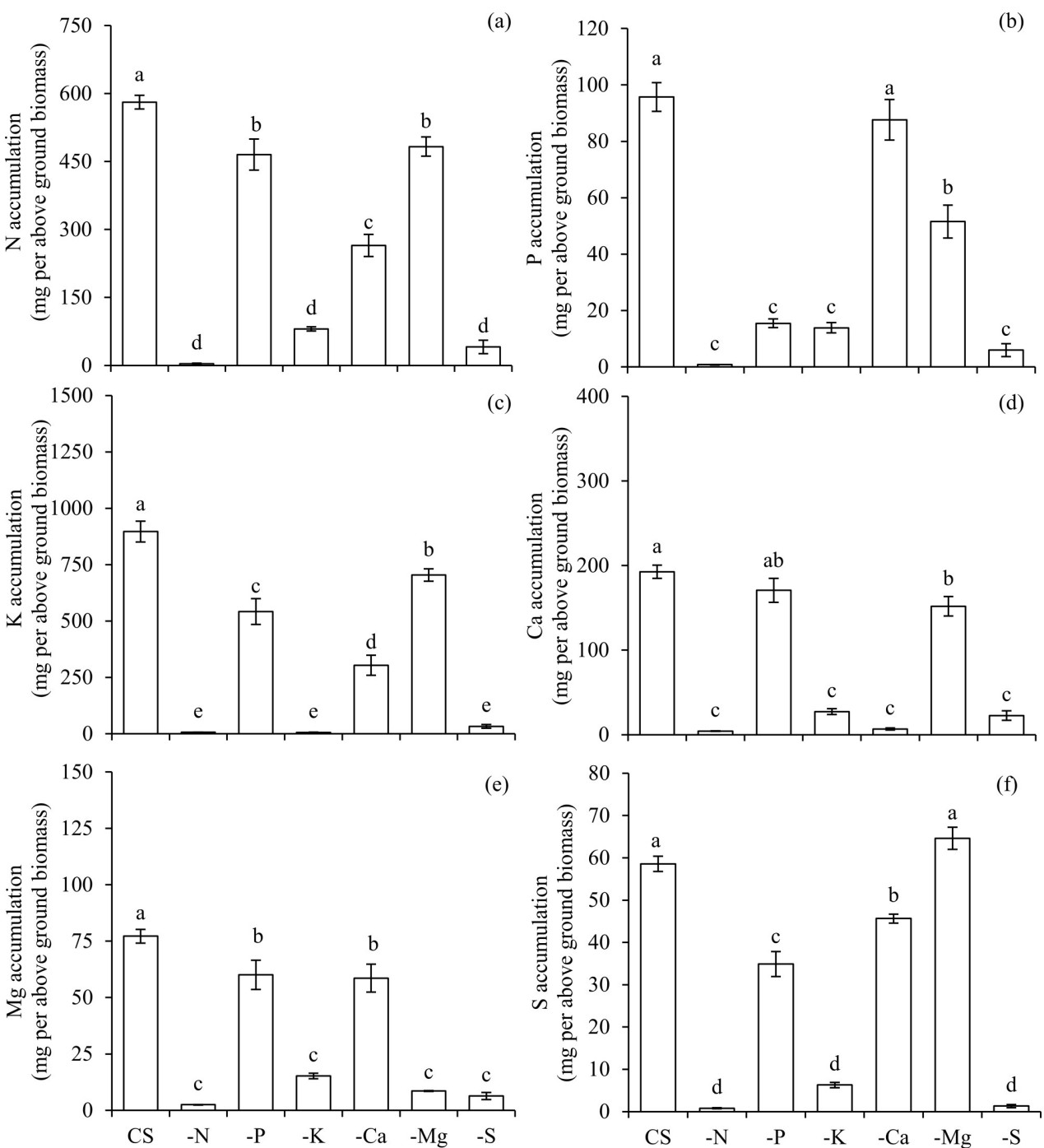

**Fig 1.** Accumulation of nitrogen (N) (a), phosphorus (P) (b), potassium (K) (c), calcium (Ca) (d), magnesium (Mg) (e), and sulphur (S) (f) in the above ground biomass of scarlet eggplants in complete solution (CS), and under the omission (-) of macronutrients (−N, −P, −K, −Ca, −Mg, and −S). Means followed by the same letter in each bar did not differ from each other by the Tukey's test (p≤0.05). Bars represent the standard error of the mean.

to other treatments (Fig 3B). However, the use efficiency of P in plants under–P was the highest among all treatments (Fig 4B). Nevertheless, the use efficiency of N (Fig 4A), K (Fig 4C), Ca (Fig 4D), Mg (Fig 4E) and S (Fig 4F) decreased in relation to CS, and for this reason, plants

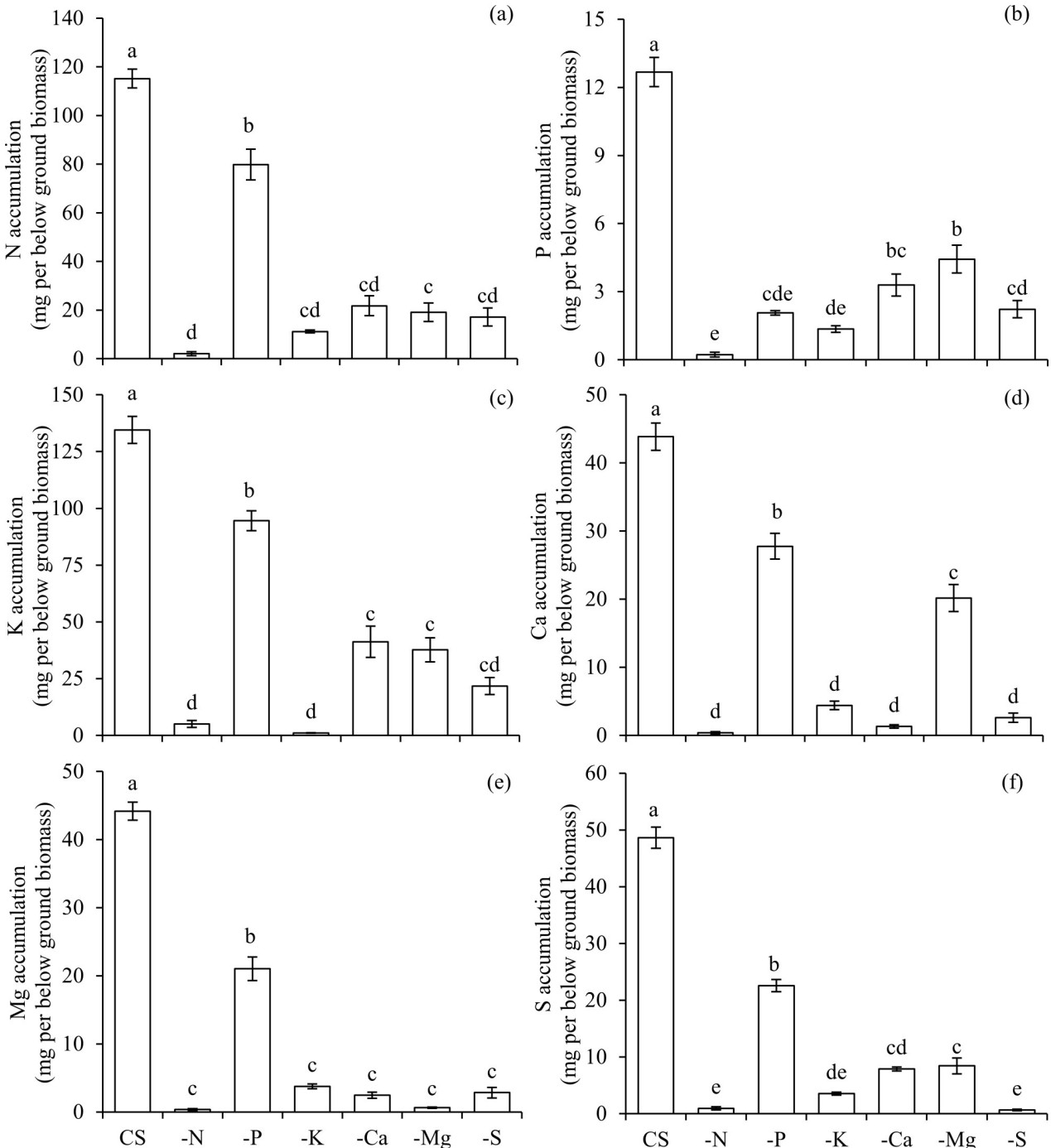

**Fig 2.** Accumulation of nitrogen (N) (a), phosphorus (P) (b), potassium (K) (c), calcium (Ca) (d), magnesium (Mg) (e), and sulphur (S) (f) in the below ground biomass of scarlet eggplants in complete solution (CS), and under the omission (-) of macronutrients (−N, −P, −K, −Ca, −Mg, and −S). Means followed by the same letter in each bar did not differ from each other by the Tukey's test (p≤0.05). Bars represent the standard error of the mean.

growth was hampered in terms of height (Fig 5A), stem diameter (Fig 5B), number of leaves (Fig 5C), leaf area (Fig 5D) and biomass accumulation in above ground (Fig 5E) and below ground (Fig 5F).

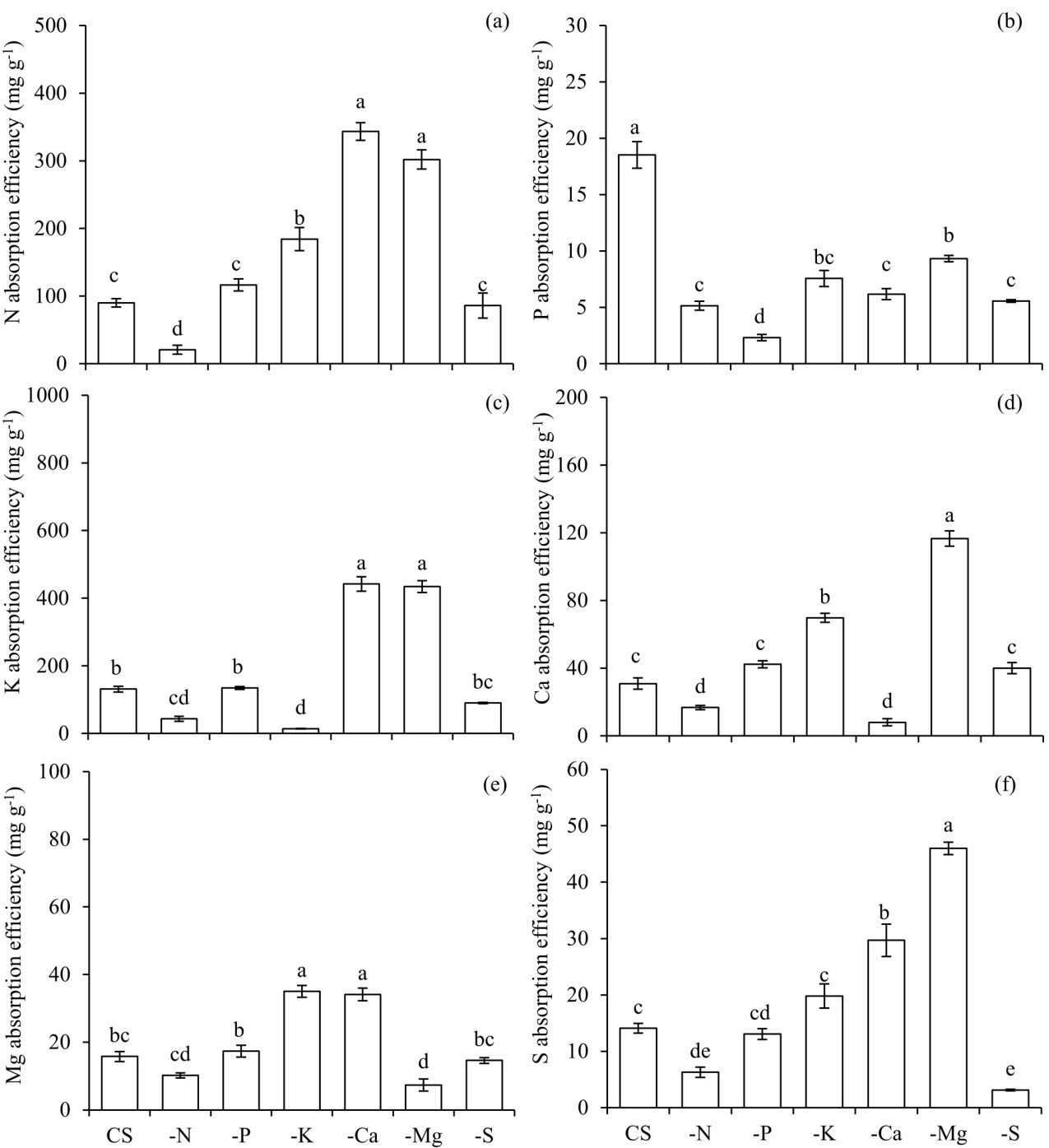

**Fig 3.** Absorption efficiencies of nitrogen (N) (a), phosphorus (P) (b), potassium (K) (c), calcium (Ca) (d), magnesium (Mg) (e), and sulfur (S) (f) in scarlet eggplants in complete solution (CS), and under the omission (-) of macronutrients (−N, −P, −K, −Ca, −Mg, and −S). Means followed by the same letter in each bar did not differ from each other by the Tukey's test (p≤0.05). Bars represent the standard error of the mean.

P was the least limiting nutrient for plants growth, with a slight reduction in biomass production in above ground and below ground of 46 and 40% in relation to CS, respectively (Fig 5E and 5F). The visual symptoms associated with P deficiency were mainly characterized by reduced height and mild chlorosis in older leaves (Figs 6B and 7B).

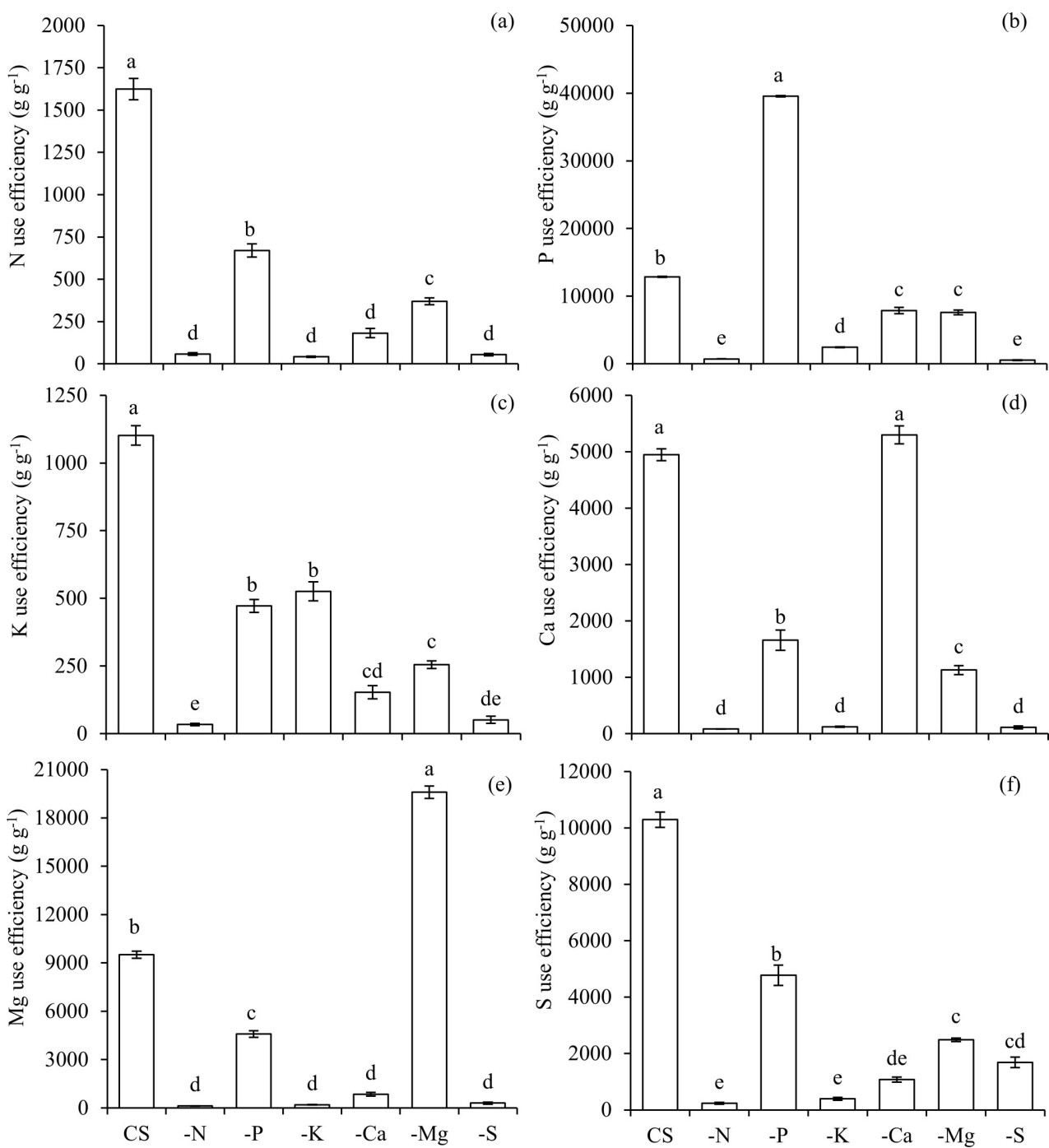

**Fig 4.** Use efficiencies of nitrogen (N) (a), phosphorus (P) (b), potassium (K) (c), calcium (Ca) (d), magnesium (Mg) (e), and sulfur (S) (f) in scarlet eggplants in complete solution (CS), and under the omission (-) of macronutrients (−N, −P, −K, −Ca, −Mg, and −S). Means followed by the same letter in each bar did not differ from each other by the Tukey's test (p≤0.05). Bars represent the standard error of the mean.

The suppression of K also led to a smaller accumulation of this nutrient in the above (Fig 1C) and below ground biomass (Fig 2C), which in turn reduced the accumulation of N (Figs 1A and 2A), P (Figs 1B and 2B), Ca (Figs 1D and 2D), Mg (Figs 1E and 2E), and S (Figs 1F and 2F) in above and below ground biomass, when compared with the CS. Plants grown in a

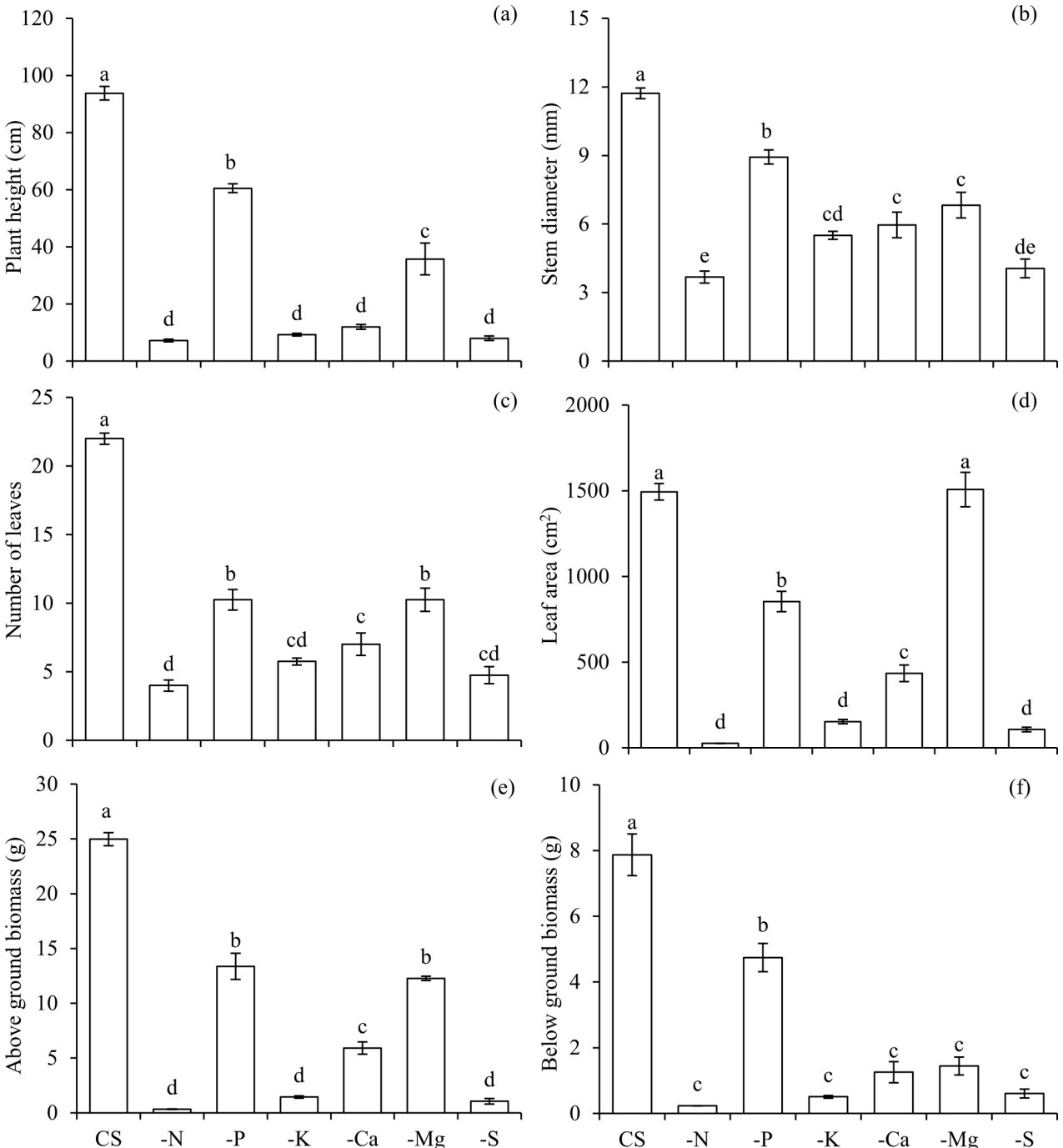

**Fig 5.** Plant height (a), stem diameter (b), number of leaves (c), leaf area (d), above ground biomass (e), and below ground biomass (f) in scarlet eggplants in complete solution (CS), and under omission (-) of macronutrients (−N, −P, −K, −Ca, −Mg, and −S). Means followed by the same letter in each bar did not differ from each other by the Tukey's test (p≤0.05). Bars represent the standard error of the mean.

condition of K deficiency reduced the absorption efficiency of this nutrient (Fig 3C), as well as of P (Fig 3B), in comparison to CS. The use efficiency of all macronutrients was reduced in the condition–K (Fig 4A–4F), in comparison to plants cultivated with CS. This nutritional imbalance caused a reduction of plants' height (Fig 5A), stem diameter (Fig 5B), number of leaves

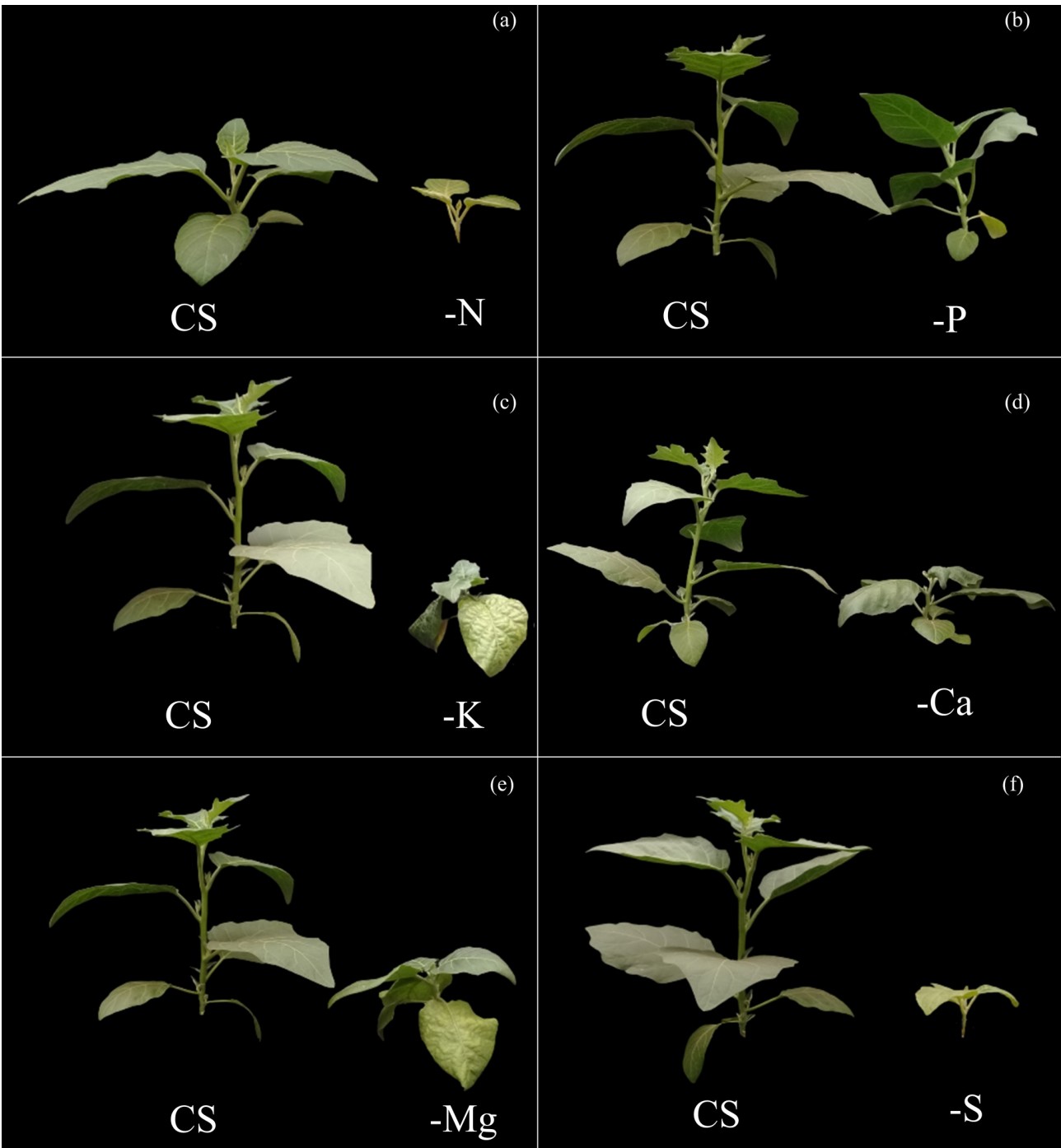

**Fig 6.** Side view of scarlet eggplants with visual symptoms of nitrogen deficiency (-N) (a), phosphorus (-P) (b), potassium (-K) (c), calcium (-Ca) (d), magnesium (-Mg) (e) and sulphur (-S) (f) compared to the complete solution (CS).

(Fig 5C), and leaf area (Fig 5D), which in turn undermined the biomass production above ground (Fig 5E) and below ground (Fig 5F).

Potassium was the third most limiting nutrient for plants growth, being responsible for losses of 94 and 93% in the biomass production in above ground and below ground in relation

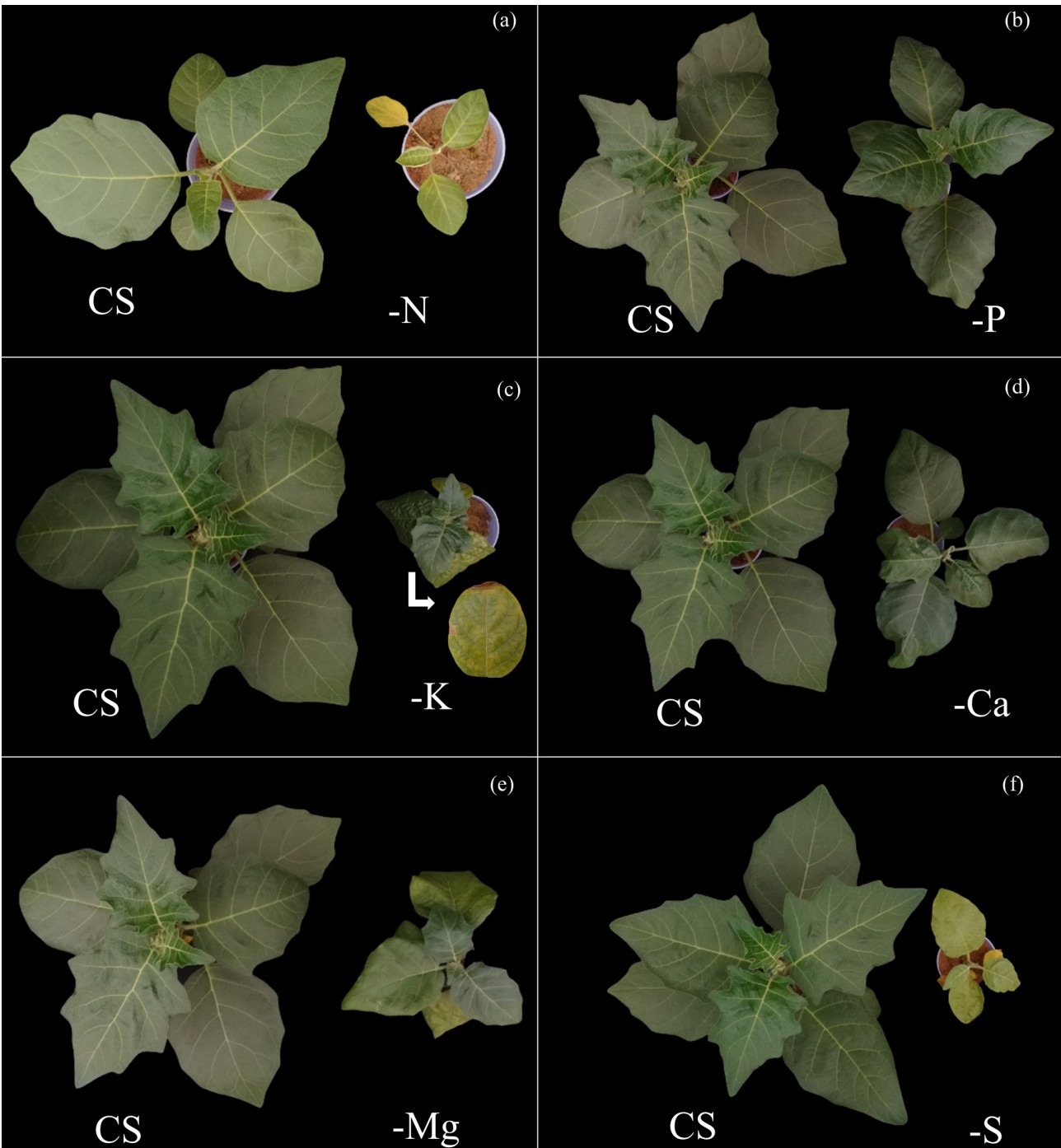

**Fig 7.** Top view of scarlet eggplants with visual symptoms of nitrogen deficiency (-N) (a), phosphorus (-P) (b), potassium (-K) (c), calcium (-Ca) (d), magnesium (-Mg) (e) and sulphur (-S) (f) compared to the complete solution (CS).

to CS, respectively (Fig 5E and 5F). The symptoms related to K deficiency were characterized by an interruption of growth, as well as irregularities in the limb of leaves and yellowing of its edges, with an advanced symptomatology revealing chlorotic spots suffering necrosis (Figs 6C and 7C).

Plants cultivated in the condition of–Ca presented a low accumulation of this nutrient in above (Fig 1D) and below ground biomass (Fig 2D). In addition, Ca deficiency reduced the absorption of N (Figs 1A and 2A), K (Figs 1C and 2C) and Mg (Figs 1E and 2E) in above and below ground biomass, as well as P (Fig 2B), and S (Fig 2F) in below ground biomass, in comparison to plants grown in CS. The absorption efficiency of Ca (Fig 3D) decreased in–Ca; however, despite the fact that this condition of Ca deficiency caused a reduced use efficiency of N, P, K, Mg and S, the use efficiency of Ca was increased, being equal to the plants cultivated in CS (Fig 4A–4F). In this sense, reductions in nutritional efficiencies caused detrimental effects in plants growth, observed in its height (Fig 5A), stem diameter (Fig 5B), number of leaves (Fig 5C), and leaf area (Fig 5D), which in turn resulted in lower biomass production in above (Fig 5E) and below ground (Fig 5F).

Ca was the fourth most limiting nutrient for plants growth, resulting in a reduced biomass production in both above ground and below ground of 76 and 84% in relation to CS, respectively (Fig 5E and 5F). The symptoms caused by Ca deficiency were initially characterized by an under development, which subsequently evolved to deformed limbs of new leaves that had irregular textures, with curved edges facing down (Figs 6D and 7D).

The deficiency of Mg was characterized by a reduction of Mg accumulated in above (Fig 1E) and below ground biomass (Fig 2E), causing an imbalance in the accumulation of other macronutrients, such as N (Figs 1A and 2A), P (Figs 1B and 2B), K (Figs 1C and 2C), and Ca (Figs 1D and 2D), in both the above and below ground biomass, and S in below ground biomass (Fig 2F), in relation to CS. Plants cultivated in the absence of Mg displayed a lower absorption of this nutrient (Fig 3E); however, the absorption efficiencies of N (Fig 3A), P (Fig 3B), K (Fig 3C), Ca (Fig 3D), and S (Fig 3F) were higher in comparison to plants grown in CS.

The use efficiency of Mg in plants cultivated under–Mg was higher in comparison to all other treatments (Fig 4E). Nevertheless, the use efficiencies of N, P, K, Ca and S were reduced in comparison to the ones observed in plants in CS (Fig 4A–4D and 4F). These effects combined reflected in reduced height (Fig 5A), stem diameter (Fig 5B), number of leaves (Fig 5C), and biomass production of above (Fig 5E) and below ground (Fig 5F), in comparison to CS. Mg was the fifth most limiting nutrient to plants development, reducing by 51 and 82% the dry matter production of above and below ground biomass in relation to CS, respectively (Fig 5E and 5F). Plants grown under Mg deficiency presented lower height and leaves of the third inferior part with irregular limb and chlorosis among veins (Figs 6E and 7E).

The absence of S in the solution reduced its accumulation in above (Fig 1F) and below ground biomass (Fig 2F). Associated with this, lower accumulations of N (Figs 1A and 2A), P (Figs 1B and 2B), K (Figs 1C and 2C), Ca (Figs 1D and 2D) and Mg (Figs 1E and 2E) were observed in above and below ground biomass, in relation to CS. The absorption efficiency of S was severely reduced in the condition of–S (Fig 3F), which was accompanied by losses in the use efficiency of all other macronutrients (Fig 4A–4F). As a reflex of damages, a reduction in plants height (Fig 5A) was observed, as well as lower stem diameter (Fig 5B), number of leaves (Fig 5C), leaf area (Fig 5D), biomass of above ground (Fig 5E) and below ground (Fig 5F).

Sulfur was the second most limiting nutrient for plants growth, leading to reductions of 96 and 92% in the production of biomass in above and below ground in relation to CS, respectively (Fig 5E and 5F). The visual symptoms of plants were evidenced initially by an interrupted growth, followed by generalized chlorosis in new leaves that evolved to other leaves (Figs 6F and 7F).

From the losses of biomass production above and below ground, the limiting order of macronutrients was N, S, K, Ca, Mg, and P, with respective reductions of 99, 96, 94, 76, 51 and 46%, in comparison to plants grown in CS (Fig 5E). Considering the losses associated with the development of below ground biomass, the limiting order of macronutrients was N, K, S, Ca,

Mg, and P, with total reductions of 97, 93, 92, 84, 82 and 40%, respectively, in comparison to plants grown in CS (Fig 5F). Thus, the omissions of N, S and K in the nutrient solution resulted in the lowest accumulation of all macronutrients in above ground biomass of plants, while the omissions of Mg and Ca caused reduced accumulations of N, Ca and Mg in above ground biomass of scarlet eggplant plants, in comparison to plants cultivated under CS condition (Fig 1A–1F).

The cluster analysis grouped the omissions of N, S and K, demonstrating that the deficiencies of these nutrients caused the greatest imbalances in the absorption of other macronutrients, with reflexes in losses of nutrients use and absorption that decreased biomass production in up to 90%. The suppression of Ca and Mg were grouped due to its similarity in limiting scarlet eggplants growth, with losses of up to 80%. However, the omission of P did not lead to significant damages in relation to plants' development. The condition of P omission was grouped with plants cultivated under CS, even though the losses related to the lack of this nutrient were around 40% (Fig 8).

## 4. Discussion

Nutritional deficiencies can limit plants growth by causing imbalances and reducing the nutrient use efficiency, which in turn decreases biomass production [6, 9]. In this study, we demonstrated that the effects caused by the lack of a nutrient ends up hampering the growth of scarlet eggplants, due to modifications in the absorption and use of the lacking nutrient, as well as other nutrients. This occurs because of the interactions between nutrients during the process of absorption and use by the plant's metabolism in converting them into dry matter [5, 8, 17].

Based on the results obtained in this study, it was evident that N was the most limiting nutrient for the growth of scarlet eggplant, due to the nutritional damages caused by this and/or other nutrients. A limited supply of N reduced the accumulation and the efficiencies of absorption and use of other nutrients, in comparison to plants that were cultivated in a condition of N sufficiency. This response occurred because the plants grown under N deficiency had a reduced accumulation of P, K, Ca, Mg and S, in both above and below ground biomass, demonstrating a severe nutritional imbalance, caused by a lower absorption efficiency of these nutrients. By participating in the constitution of enzymes and cell membrane transporters in roots [6], N is involved in the process of active absorption of nutrients, thus its deficiency affects this process negatively. Associated with this, the nutritional imbalances caused by N deficiency reduced its use efficiency, which was accompanied by the ones of P, K, Ca, Mg and S.

The biological functions performed by nutrients in plants induce interactions between multiple nutrients [9], and due to this reason, the nutritional deficiency of N compromises the metabolism of other nutrients, even if those are not lacking in the nutrient solution. The low use efficiency of S as a result of N shortage is due to the participation of both nutrients in the protein synthesis, seen that both are constituents of essential amino acids [18]. In addition, low concentrations of N in leaves result in a reduced production of photosynthetic pigments [19], which ends up affecting the metabolism of Mg, as both constitute the structure of chlorophyll molecules that are essential for the absorption of photons and for the transportation of electrons during photosynthesis [20].

This disturbance caused by N deficiency in electron transportation affects the whole photosynthetic metabolism by reducing the use efficiency of P, seen that this nutrient acts in several metabolic activities related to photosynthesis and in the activity of 1,5-biphosphate carboxylase ribulose (Rubisco) [21]. When these factors are combined, the enzymes involved in both protein synthesis and photosynthesis have their activity reduced, because N deficiency also

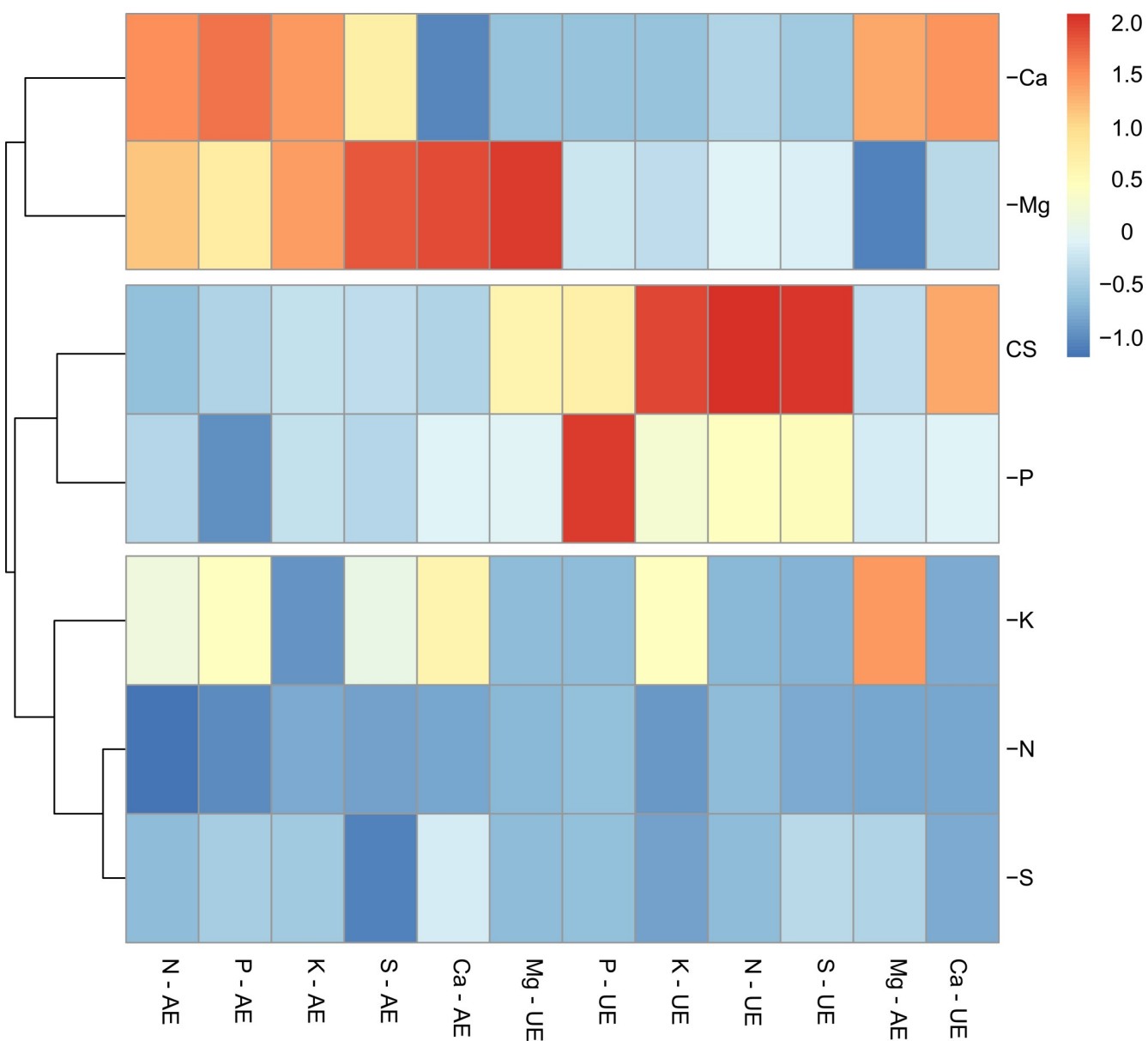

**Fig 8. Hierarchical cluster analysis with standardized data on absorption efficiencies (EA) and use (EU) of macronutrients in scarlet eggplants under nitrogen (-N), phosphorus (-P), potassium (-K), calcium deficiency (-Ca), magnesium (-Mg) and sulfur (-S) and complete solution (CS).**

compromises the absorption of K, preventing them to be activated, given the role of K in the enzymatic activity [22]. In this sense, N deficiency reduced growth and biomass production of plants by affecting several physiological mechanisms that depend on nutritional homeostasis. These events culminated in the appearance of visual symptoms in the foliar tissue, which consisted of generalized chlorosis in old leaves that over time progressed to all leaves, in a similar way as described by Haag et al. [11] in scarlet eggplants.

The second nutrient that limited the most the development of plants was S, and the damages caused by its omission were exacerbated because plants cultivated under S deficiency accumulated less of this and other nutrients. It was evidenced that S deficiency compromised the absorption of this nutrient, but did not affect the absorption efficiency of other nutrients;

however, the use efficiency of S was reduced by 84% in comparison to plants cultivated in CS, beings accompanied by reductions in the use efficiencies of other macronutrients. This result demonstrates that the growth of scarlet eggplants is more affected by reductions in the nutrients use efficiency, in comparison to absorption efficiencies.

The reduced use efficiency of N as a result of S deficiency occurred for the same reason described in the condition of–N, i.e. the participation of both nutrients in the protein synthesis, because both are constituents of essential amino acids [18]; therefore, S deficiency also impaired the functions performed by N. This deficiency also affected the metabolism of Mg, possibly due to the participation of both nutrients in the formation of photosynthetic pigments, seen that chloroplasts' tilacoidal membranes are constituted by sulpholipid compounds [23]. Thus, with a reduction of these compounds, chlorophyll formation is inhibited and the metabolism of Mg is impaired [24]. Additionally, Mg acts together with P in the assimilatory reduction of $SO_3^{-2}$ [4], and for this reason, the depletion of S affects these nutrients' metabolism. Still, S acts plays an important role in the composition of ferredoxins, which are enzymatic complexes involved in photosynthesis [4]; thus it is another path that may justify the damages caused by P metabolism in plants cultivated under S deficiency.

Sulphur deficiency also resulted in a reduced use efficiency of K, seen that both the loading of K in the plant's xylem and its translocation to the aerial part are partially determined by the amount of sulphate translocated, and accumulated in the aerial part of the organism, due to its action as counter-ion [12]. Therefore, K accumulates in roots when sulphate is lacking in the plant. This effect was evidenced in this study, in which 40% of the K in plants cultivated under–S was accumulated in below ground biomass, reaching a value about twice higher in comparison to plants cultivated in CS.

The nutritional imbalance caused by S deficiency compromised the role of Ca in plants, which is involved in the formation of the tissues' cell well [4], also affecting its metabolism, as observed by the lower use efficiency of this nutrient and consequently in the formation of new leaves. Such nutritional imbalances led to a reduced growth of scarlet eggplants, and induced visible symptoms such as uniform chlorosis of new leaves.

It was possible to verify that K was the third most limiting nutrient in relation to biomass accumulation and growth of scarlet eggplants, because it also affected the nutritional homeostasis and reduced the concentration of all other nutrients in the plants. However, the absorption efficiency of N, P, Ca, and Mg was augmented in comparison to plants grown in CS, demonstrating that this species has nutritional mechanisms that interact in order to reduce the damages caused by K deficiency. It is currently known that plants cultivated in environments with deprivation of K might use the mechanism of absorption by transporters with high affinity by $K^+$, which involves high-energy expenditures (ATP) [10], thus justifying the increased absorption efficiency of P in plants deficiency in K, as observed in this study. In addition, plants may also stimulate the influx of $K^+$ using $NO_3^-$ as counter-ion for electrical balance, a fact that was also observed in this study by the increased absorption efficiency of N, which corroborates the findings of Kellermeier et al. [25]. The role of $K^+$ in osmotic functions may also be partially replaced by other cations, such as $Mg^{2+}$ or $Ca^{2+}$ [7], and there is evidence that plants cultivated under $K^+$ restriction tend to absorb more of these cations in order to maintain the cellular ionic equilibrium [12, 26]. These events contribute to justify the increased absorption efficiency of Ca and Mg by scarlet eggplants under K deficiency.

The scenario of potassic deficiency caused prejudices in the metabolism of several nutrients because a reduction of 52% in the use efficiency of K was observed, which was accompanied by reductions in the use of N, P, Ca, Mg and S, all of approximately 97% in comparison to plants cultivated in CS. The lowest concentration of K in the foliar tissue causes reductions in the synthesis of proteins and in the accumulation of soluble nitrogen compounds, with putrescine, N-

carbamoyl putrescine and agmatine [27]. This effect reflects in a lower use efficiency of N and K, seen that these nutrients are involved in the synthesis of proteins and interfere in the biological function of other nutrients, seen that putrescine is toxic to plants and causes necrosis in tissue. In addition, K deficiency induces the production of ethylene, which in turn regulates positively the production of oxygen reactive species in roots [10], inducing oxidative stress and interfering in vital metabolisms of plants. This nutrient is also strongly linked to the loading of photo-assimilated compounds in the phloem, to the maintenance of the membrane potential and activation of different enzymes related to the metabolism of photosynthesis [28]. In this sense, the low absorption of K also caused a reduced efficiency of the use of N, P, Ca, Mg and S, by the function that this nutrient exerts in the enzymatic processes and in the plant's metabolism.

When facing nutritional disturbances, plants that were deficient in K displayed visual symptoms characterized by marginal chlorosis in old leaves that evolved to necrosis, because of the accumulation of putrescine in this region. Associated with this, K deficiency was characterized by irregularities in the leaf limb of old leaves, similar to the results reported by Haag et al. [11]; however, those authors did not describe chlorosis in the edges of leaves, as observed in the present study.

Ca deficiency resulted in a nutritional imbalance in the plant by reducing the accumulation of Ca, K, N, Mg and S. In addition, it was evidenced that the absorption efficiency of Ca decreased, while the ones of N, K and Mg increased in comparison to plants cultivated in CS. Similarly, to the results observed in plants grown under–K, we believe that plants under Ca deficiency have increased the absorption of cations such as $K^+$ and $Mg^{2+}$, boosted by a charge equilibrium [12, 26]. The increased efficiency of N absorption was due to the use of ammonium-N in plants grown with the omission of Ca, as indicated by Hoagland and Arnon [13]. Therefore, the high efficiency of ammonium transporters [29] contributed for the increased efficiency of N absorption by plants, which in turn reflected in the efficiency of S absorption, seen that N is a constituent of the enzymes that transport all nutrients, including S [4].

The deficiency of Ca did not interfere in the use efficiency of this macronutrient, being similar to the ones observed in plants cultivated in CS. This can be attributed to the function of Ca, which is stimulated only in low concentrations of the element in the cellular cytosol, acting as a signal for photosynthesis-related enzymes, and as a secondary messenger in abiotic stress events [30], such as the ones caused by nutritional deficiencies. In addition, the plants received Ca throughout its adaptation stage, which may have been accounted in the chemical analysis performed in its aerial parts. Nevertheless, Ca was the fourth most limiting nutrient for the growth of scarlet eggplants in this experiment. This fact occurred because Ca deficiency caused an imbalance in the metabolism of other nutrients, reducing in 93, 91, 90, 89 and 86% the use efficiencies of P, Mg, S, N and K, respectively.

The low efficiencies of use of other macronutrients in an environment deficient in Ca were due to disturbances in the stability of the cytosol structure [31]. Ca deficiency causes cell wall malformation, seen that this structure is maintained by $Ca^{+2}$ bonds to pectates [30] that reflects in the formation of organelles such as mitochondria, which play an important role in the accumulation of ions [31]; thus, even though ions were absorbed, its incorporation in the plants metabolism was hampered. The nutritional disturbances caused by Ca deficiency evolved and reached tissue level, causing visual symptoms that start in young leaves with curved margins facing down due to cells' malformation, similarly to the symptoms reported by Haag et al. [11].

Mg was the fifth most limiting macronutrient for the development of plants, and its deficiency reduced the accumulation of N, P, K, Ca and Mg, both in above and below ground biomass, and S in below ground biomass, in comparison to CS. Nevertheless, the absorption efficiency of these macronutrients was higher in plants under Mg deficiency, which

demonstrated that nutritional interactions can be strategic to reduce the impacts caused by Mg deficiency in scarlet eggplants, seen that in this condition, plants tend to increase P absorption, in order to maintain the activation of enzymes in photosynthetic processes, such as the Calvin cycle, seen that the lack of Mg reduced the reaction speed and the affinity between the enzyme Rubisco and the substrate [4]. Additionally, low concentrations of Mg in the nutrient solution stimulated the absorption of cations (e.g. $K^+$ and $Ca^{2+}$), to maintain the equilibrium of charges [12]. The increased efficiency of N absorption in plants deficient in Mg can also be attributed to the use of a source of ammonium-N, inducing an increased absorption efficiency of N because of the high affinity of ammonium transporters [29].

It was shown that the strategy of plants grown under Mg deficiency is not increasing its absorption efficiency, but in the use efficiency of this macronutrient, which even surpassed other nutrients. This was due to a greater internal cycling because plants redistributed the Mg that was linked to molecules, and stored it in cellular vacuoles of old leaves, so it could be transferred to vascular tissues and reach new leaves, ensuring Mg homeostasis [32]. Nonetheless, the damages caused by Mg deficiency impaired the metabolism of other nutrients, reducing the use efficiency of N, P, K, Ca and S.

The lowest use efficiency of N in plants deficient in Mg was due to the participation of Mg in the stability of ribosomes, by keeping the subunits linked to amino acids and mRNA, guaranteeing the effectivity of the protein synthesis [4]. In this sense, plants deficient in Mg accumulated non-protein N, reducing the use efficiency of N. In addition, N and Mg as basic components of chlorophyll molecules, and around 10–20% of the Mg present in plants are linked to these pigments [24]. Thus, this could be one of the mechanisms by which the suppression of Mg interferes in N metabolism. Furthermore, we believe that the reduced formation of these pigments has contributed for the reduction of the S use efficiency, given the participation of this nutrient in the constitution of sulpholipid compounds in the membranes of photosynthetic pigments [23].

The decreased chlorophyll content in plants deficient in Mg was also due to the degradation caused by sugars and starch accumulation in the cells of deficient leaves [33], which caused a super-reduction of the transport chain of photosynthetic electrons, generating reactive oxygen species that degrade these pigments [34]. In this sense, the resulting oxidative stress may have caused negative effects in Ca metabolism, as reactive species cause damages to the cell wall [35]. In addition, Mg accompanies Ca in the formation of pectin in the cell wall [4], demonstrating that Mg deficiency limits Ca metabolism by reducing the formation and increase the degradation of the cell wall.

The reduction of photosynthetic pigments content caused by the condition of Mg deficiency disturbed the transportation of electrons, compromising the photosynthetic system [20], and consequently the metabolism of P. When associated to the participation of Mg as a cofactor of a series of enzymes involved in the photosynthetic fixation of carbon [33], those events demonstrate that the reactions triggered by Mg reduction in the cell content reflect in the use efficiency of P by plants. In addition, Mg deficiency generates an inhibition in sucrose loading in the phloem, seen that this process is catalyzed by a cotransporter $H^+$/sucrose, which its activity requires a gradient of protons and is maintained by a $H^+$-ATPase located in the plasmatic membranes of the cells in the vascular system. Evidence suggests that Mg-ATP is an important ATP complex in cells, essential to the activity of $H^+$-ATPase [33]. This accumulation of sugars has also impaired the metabolism of K.

The symptoms of Mg deficiency were initially characterized by an appearance of an internerve chlorosis in older leaves, which occurs by low concentrations of this nutrient in the foliar limb. The symptomatology described in this study are similar to the ones previously reported in scarlet eggplants [11].

In the experimental conditions of this study, even though P deficiency harmed plants in comparison to plants cultivated in CS, P was the macronutrient with the lowest interference in plants growth, which can also be attributed to an increased efficiency of this nutrient. The high use efficiency of P by plants deficient in this nutrient occurred because of a reduced structure of tilacoidal membranes of chloroplasts, which induce an increased capture of photons and consequently in the photosystem II efficiency [36]. Associated to this, the suppression of P maintained the use efficiencies of Ca, N, K, S and Mg at a moderate level, with reductions of 66, 59, 57, 54 and 52%, respectively, in comparison to plants cultivated in CS. In this sense, plants that were deficient in P presented higher metabolic activity, especially regarding enzymatic activities, and had reduced losses in the protein synthesis, which in turn caused a lower effect in the metabolism of N, S and K, as well as in the formation of new tissues without affecting Ca metabolism. In spite of this, P suppression reduced the use efficiency of Mg, due to the low demand of Mg in enzyme activation [4]. Thus, plants submitted to P deficiency presented a later symptomatology, characterized by a darker green color in young leaves, and yellowish tones on the lower third of old leaves. These results corroborate the ones presented by Haag et al. [11] in scarlet eggplants.

The results presented in this study support the hypothesis that a macronutrient's deficiency modifies both the absorption and use efficiency of other nutrients, causing a series of biological signaling events that are related to the adaptive responses of plants, whose integration interfere in the growth and development of scarlet eggplants. The study proposes in an unprecedented way that future research evaluating the biological damages caused by the lack of a certain nutrient should take into account other nutrients as well, even if these are found in adequate concentrations in the nutrient solution, in order to better understand the extension of nutritional damages in the growth of plants.

## 5. Conclusion

The biological damages caused by nutritional deficiency in scarlet eggplant depend on the nutrient omitted in the nutrient solution associated with its interaction with other nutrients, as it affects the absorption efficiency and use efficiency of these elements by plants. The most limiting nutrients were N, S and K, seen that their deficiencies resulted in deleterious effects in the metabolism of all other nutrients, demonstrating the importance of an adequate nutritional management in scarlet eggplant crops.

## Supporting information

**S1 Data. This file contains all data the plants of experiment.**
(XLSX)

## Author Contributions

**Conceptualization:** Gelza Carliane Marques Teixeira, Renato de Mello Prado, Kamilla Silva Oliveira, Antonio Carlos Buchelt, Antonio Márcio Souza Rocha.

**Data curation:** Gelza Carliane Marques Teixeira, Antonio Márcio Souza Rocha.

**Formal analysis:** Gelza Carliane Marques Teixeira, Kamilla Silva Oliveira, Antonio Carlos Buchelt, Antonio Márcio Souza Rocha, Michelle de Souza Santos.

**Funding acquisition:** Renato de Mello Prado.

**Investigation:** Gelza Carliane Marques Teixeira, Renato de Mello Prado, Kamilla Silva Oliveira, Antonio Carlos Buchelt, Antonio Márcio Souza Rocha.

**Methodology:** Gelza Carliane Marques Teixeira, Renato de Mello Prado, Kamilla Silva Oliveira, Antonio Carlos Buchelt, Antonio Márcio Souza Rocha, Michelle de Souza Santos.

**Project administration:** Renato de Mello Prado.

**Resources:** Renato de Mello Prado.

**Supervision:** Renato de Mello Prado.

**Validation:** Gelza Carliane Marques Teixeira, Renato de Mello Prado, Kamilla Silva Oliveira, Antonio Carlos Buchelt, Antonio Márcio Souza Rocha.

**Writing – original draft:** Gelza Carliane Marques Teixeira, Kamilla Silva Oliveira, Antonio Carlos Buchelt, Antonio Márcio Souza Rocha, Michelle de Souza Santos.

**Writing – review & editing:** Gelza Carliane Marques Teixeira, Renato de Mello Prado.

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
