## [Decision Letter · Decision Letter 0]

8 Apr 2021

PONE-D-21-00622

Nutritional deficiency in scarlet eggplant limits its growth by modifying the absorption and use efficiency of macronutrients

PLOS ONE

Dear Dr. Teixeira,

Thank you for submitting your manuscript to PLOS ONE. After careful consideration, we feel that it has merit but does not fully meet PLOS ONE’s publication criteria as it currently stands. Therefore, we invite you to submit a revised version of the manuscript that addresses the points raised during the review process.

We look forward to receiving your revised manuscript.

Kind regards,

Umakanta Sarker

Academic Editor

PLOS ONE

Journal Requirements:

Reviewers' comments:

Reviewer's Responses to Questions

**Comments to the Author**

1. Is the manuscript technically sound, and do the data support the conclusions?

Reviewer #1: Partly

Reviewer #2: Yes

2. Has the statistical analysis been performed appropriately and rigorously? 

Reviewer #1: Yes

Reviewer #2: Yes

3. Have the authors made all data underlying the findings in their manuscript fully available?

Reviewer #1: Yes

Reviewer #2: Yes

4. Is the manuscript presented in an intelligible fashion and written in standard English?

Reviewer #1: No

Reviewer #2: No

5. Review Comments to the Author

Reviewer #1: Abstract

Line 30. Accumulation of what? Please specify.

Lines 39 – 41. I would recommend replacing the phrases “aerial part” and “root” to “above ground” and “below ground” biomass respectively.

Line 41-45. The order of limitation of macronutrients presented by the authors is contrary to the interpretation presented. The authors claimed that the order of limitation was N<s<k<ca<mg<p. and="" authors="" be="" but="" claimed="" least="" limiting="" most="" n.="" nutrient="" order="" p="" that="" the="" this="" would="">Introduction

Line 53 – 56. Too long sentence please consider breaking into two

Line 54. Change the word this sentence “plant nutritional is a complex factor, seen that in reality there are interactions between nutrients” to “plant nutrition is a complex factor, due to the interactions between nutrients”

Line 57. “Such deficiencies can cause a nutritional imbalance” please remove “a”

Line 60. Please consider changing “agronomical interest” to “agronomic potential”

Line 65 - 67. “This occurrence may hamper the metabolism of other nutrient, such as N, which has as one of its noble functions the increment of protein synthesis in the plant, but this function is in turn interrupted due to the low activity of enzymes involved in the assimilation of N.

There are multiple grammatical and word-use errors in this sentence please reword.

The authors mentioned that the scarlet eggplant is of high agricultural interest however the importance of the crop to the region of study was not included in the manuscript. Information such as total planted area, economic value etc will help the readers understand the importance of this study as well as the impact on the local agricultural industry.

Materials and methods

Line 90: Please change “There was a high variation in the” to “There were high variations……”

There seem to an important level of grammatical and word use error in the current version of the manuscript. Some have been highlighted; however, the authors are invited to thoroughly checked the manuscript for these errors to improve on the readability and flow of the manuscript.

Line 123. Change “until constant weight, and then weighted” to until a constant weight was obtained”

Lines 124 - 128: “The levels of N, P, K, Ca, Mg and S were determined from the digestion of the plant material samples, according to the methods described by Bataglia et al. [10], and based on the dry matter content, the concentrations of each nutrient in the plants were calculated. In addition, the efficiency of absorption and use of nutrients was calculated, using the data of macronutrients accumulation and dry matter, as recommended by Fageria and Baligar [11]”

Both nutrient content in plant biomass, absorption and nutrient use efficiencies are the most important aspect of this study therefore, other than just the reference, a detail procedure of how these parameters were estimated should be included in this manuscript.

How long did the experiment lasted? How many weeks? This information should be included in the manuscript.

Results

Line 144: Please state those nutrients that were reduced here in this section

Line 147 - 149: Representing nutrient accumulation in “mg shoot-1” is not a scientifically accepted unit, it is arbitrary and does not provide a standard way of comparison with the results from other studies. The authors are invited present their results with standardized unit.

Lines 158 -159. “The omission of N reduced the accumulation of this nutrient in the aerial parts (Fig 1a) and roots (Fig 2a)”. Same issue here, please state those nutrients here.

Line 223: Smaller accumulation compared to what?

The presentation of the results in the results section is confusing. The authors indicated percentage reduction in plant biomass for each omitted element, how the percentage reduction was estimated was not clear. What is the reference to which all the treatments were compared to?

Was there any data on the effects on yield? What would be the effects of each deficient nutrient on yield? Since the fruit is most important aspects of this crop, therefore this information would go a long was to really quantify the impact of the study on the local agricultural sector.

Figures

The unit presented are not acceptable in any scientific article. Please consider recalculating these results using a more standardized unit. A possible alternative could be mg/kg of root, shoot etc</s<k<ca<mg<p.>

Reviewer #2: The study is carefully concepted and methodological approach is satisfactory explained. The manuscript deals with interesting and important changes in response to macronutrient deficiency in scarlet eggplant.

Selected parameters are wisely chosen, all of the important nutrients are there, from nitrogen to sulfur. Novelty can be presented from the aspect that such an investigation has never been performed in scarlet eggplant. Results can help to understand the nutritional mechanisms in scarlet eggplant and other plants.

Results and discussion portions are well written. The authors also draw an accurate picture from the results. However, the authors should review and cite some more relevant references. Also needs a proper revision of English language

Many sentences are very confusing. English language and writing style needs to be improved sufficiently

Line: 25- 27. Revise the sentence “but its effects on a plant’s nutritional efficiency to absorb and utilize the missing nutrient, and the reason why these damages reflect in other nutrients have not yet been reported in the culture of scarlet eggplant”

Line 46- 47. Do not repeat words of the title in the Keywords

Line: 57-58. Add at least 2 more citation to this line. As many manuscripts have studied plants from different angles, you can cite them here “ Such deficiencies can cause a nutritional imbalance, leading to tissues deformities, reduced leaf area, limited growth and dry matter production [4]”.

Line: 63-70. As K and N are studied extensively, and many high quality articles are available about this. So, add at least 2 more references here.

Line: 108-109. “Its pH was adjusted to 5.5 ± 0.2 using solutions of hydrochloric acid (HCl) or sodium hydroxide (NaOH), both at 1.0 mol/L” replace this sentence by “Its pH was maintained between 5.3-5.7”. Omit the line “using solutions of hydrochloric acid (HCl) or sodium hydroxide (NaOH), both at 1.0 mol/L”.

Line: 110-111. Omit the “containing all nutrients”. As you have mentioned complete nutrient solution.

Line: 111-113. Mention the particular time (in the form of days) for every nutrient when the symptoms appears.

Line: 120-121. All plant materials were washed in tap water, replace with “All plant materials were washed with the tap water

Line: 124-125. It’s better to highlight the instrument used for the ionic analysis “The levels of N, P, K, Ca, Mg and S were determined from the digestion of the plant material 125 samples, according to the methods described by Bataglia et al. [10]”

Line: 152. Omit the “of Hoagland and Arnon (1950)”, as you have mentioned it in the methodology section. Also follow for all other graphs.

Line: 161-162. Replaced the sentence “ both aerial parts and roots, in comparison to the plants cultivated in the condition of CS” by “both aerial parts and roots, in comparison to the plants under all other conditions.

Line: 226. Replace “when compared to CS” with “when compared with the CS”

Line: 304-405. Please add reference to this statement.

Line: 449. If I am not wrong you want to talk about “mRNA” ? You mentioned “RNAm”. If this is the case, so revise it please.

6. PLOS authors have the option to publish the peer review history of their article (what does this mean?). If published, this will include your full peer review and any attached files.

Reviewer #1: No

Reviewer #2: **Yes: **Sunjeet Kumar

---

## [Author Response · Author response to Decision Letter 0]

14 May 2021

To

Plos One 

Manuscript: PONE-D-21-00622

Title: Nutritional deficiency in scarlet eggplant limits its growth by modifying the absorption and use efficiency of macronutrients

Abbreviations: L= line; A= Answer

General editor comments: Thank you for submitting your manuscript to PLOS ONE. After careful consideration, we feel that it has merit but does not fully meet PLOS ONE’s publication criteria as it currently stands. Therefore, we invite you to submit a revised version of the manuscript that addresses the points raised during the review process.

Answer (A): Thank you for allowing us to re-submit the revised version of the manuscript. We are very grateful for the valuable suggestions from editor and for spending a considerable amount of time reading it. The editor's suggestions were appreciated by the authors and followed. Below we list the editor's suggestions with brief comments from the authors. We believe that this revised version is better and in accordance to the editor comments.

Reviewer #1: 

Abstract

Line 30. Accumulation of what? Please specify.

A: The accumulation we refer to is that of macronutrients. We have included this information in text (L 30).

Lines 39 – 41. I would recommend replacing the phrases “aerial part” and “root” to “above ground” and “below ground” biomass respectively.

A: We are grateful for your suggestion and we reiterate the change in the term (L 42). In order for the manuscript to be coherent with the abstract, we made this modification in figures, in legends of the figures and in entire manuscript according to its recommendation.

Line 41-45. The order of limitation of macronutrients presented by the authors is contrary to the interpretation presented. The authors claimed that the order of limitation was N

A: We understand the indication of the rapporteur and there was a failure of the authors. We chose to remove the symbol of greater or lesser and we included percentage values to describe the order of limitation (see L 43 to L 44; L 295; L 297 to L 298).

Introduction

Line 53 – 56. Too long sentence please consider breaking into two

A: The rapporteur is right. We divided the sentence in two according to the rapporteur's suggestion (L 5 to L 62).

Line 54. Change the word this sentence “plant nutritional is a complex factor, seen that in reality there are interactions between nutrients” to “plant nutrition is a complex factor, due to the interactions between nutrients”

A: The rapporteur's suggestion is relevant and important. We made the change to the text (L 59).

Line 57. “Such deficiencies can cause a nutritional imbalance” please remove “a”

A: We remove the “a” from the sentence (L 62).

Line 60. Please consider changing “agronomical interest” to “agronomic potential”

A: Great suggestion from the rapporteur. We accept the modification and make it in the text (L 65).

Line 65 - 67. “This occurrence may hamper the metabolism of other nutrient, such as N, which has as one of its noble functions the increment of protein synthesis in the plant, but this function is in turn interrupted due to the low activity of enzymes involved in the assimilation of N.

There are multiple grammatical and word-use errors in this sentence please reword.

The authors mentioned that the scarlet eggplant is of high agricultural interest however the importance of the crop to the region of study was not included in the manuscript. Information such as total planted area, economic value etc will help the readers understand the importance of this study as well as the impact on the local agricultural industry.

A: The rapporteur's suggestion is pertinent. The sentence was reformulated and grammatical errors were corrected (L 71 to L 73). In addition, we inform that the entire text has been revised and the necessary grammatical corrections have been made.

We also included in the introduction section, a paragraph that demonstrates the economic importance of jiló, citing the planted area, production, productivity and income generated from the commercialization of this vegetable in the State of São Paulo, where the work was carried out (L 54 to L 57).

Materials and methods

Line 90: Please change “There was a high variation in the” to “There were high variations……”

There seem to an important level of grammatical and word use error in the current version of the manuscript. Some have been highlighted; however, the authors are invited to thoroughly checked the manuscript for these errors to improve on the readability and flow of the manuscript.

A: We carried out the modification suggested in L 90 (currently L 96). In addition, we are grateful for the reviewer's concern to improve the quality of the writing in the text and recognize the importance of conducting a thorough review of this manuscript by an expert native to the English language, thus, the entire manuscript was subjected to grammatical corrections.

Line 123. Change “until constant weight, and then weighted” to until a constant weight was obtained”

A: The suggestion was answered and we incorporated this change into the text (L 132).

Lines 124 - 128: “The levels of N, P, K, Ca, Mg and S were determined from the digestion of the plant material samples, according to the methods described by Bataglia et al. [10], and based on the dry matter content, the concentrations of each nutrient in the plants were calculated. In addition, the efficiency of absorption and use of nutrients was calculated, using the data of macronutrients accumulation and dry matter, as recommended by Fageria and Baligar [11]”

Both nutrient content in plant biomass, absorption and nutrient use efficiencies are the most important aspect of this study therefore, other than just the reference, a detail procedure of how these parameters were estimated should be included in this manuscript.

How long did the experiment lasted? How many weeks? This information should be included in the manuscript.

A: The reviewer is right and we agree that the incorporation of further descriptions of these methods is important, as follows: 

“The levels of P, K, Ca, Mg, and S were determined by the digestion of samples, using a digestive mixture of perchloric and nitric acid (1:2), with readings of K, Ca, and Mg performed in spectrophotometry of atomic absorption with air-acetylene flame, while P and S readings were carried out by means of spectrophotometry [14]. The concentration of each nutrient was calculated based on the plant biomass. 

In addition, the absorption and use efficiency of nutrients were calculated using the data of macronutrients accumulation and plant biomass, as recommended by Fageria and Baligar [15]. For this purpose, distinct equations were used to calculate the absorption efficiency: (accumulation in the whole plant/biomass of root); and use efficiency: ((biomass of whole plant)2/accumulation in the whole plant).” (L 133 to L 144).

In addition, we have included more information about the duration of the experiment (L 96), as well as the time of onset of visual deficiency symptoms of each omitted nutrient that corresponds to the collection date of each treatment (L 119 to L 121). 

Results

Line 144: Please state those nutrients that were reduced here in this section

A: We specified which nutrients were reduced and this increased the clarity of the sentence (L 159 to L 160).

Line 147 - 149: Representing nutrient accumulation in “mg shoot-1” is not a scientifically accepted unit, it is arbitrary and does not provide a standard way of comparison with the results from other studies. The authors are invited present their results with standardized unit.

A: The rapporteur is right that there is no "mg shoot-1" unit foreseen in the international system of units (SI).

Therefore, this representation is not in the SI and its comparison is limited. What we need to indicate is what the result represents. There was a flaw in us indicating "mg shoot-1" that could mistakenly suggest an SI unit. To avoid this the correct thing is to use "mg per aerial part" so without "-1" notation it indicates that it is not a SI unit. This way of expressing this result that we use in the revised version is widely used in the literature (links below papers). The figures have been corrected.

de Souza Junior JP, Prado RdM, de Morais TcB, Frazão JJ, dos Santos Sarah MM, de Oliveira KR, et al. (2021) Silicon fertigation and salicylic acid foliar spraying mitigate ammonium deficiency and toxicity in Eucalyptus spp. clonal seedlings. PLoS ONE 16(4): e0250436. https://doi.org/10.1371/journal.pone.0250436

da Silva, D.L., de Mello Prado, R., Tenesaca, L.F.L. et al. Silicon attenuates calcium deficiency by increasing ascorbic acid content, growth and quality of cabbage leaves. Sci Rep 11, 1770 (2021). https://doi.org/10.1038/s41598-020-80934-6

Frazão, J.J., Prado, R.d., de Souza Júnior, J.P. et al. Silicon changes C:N:P stoichiometry of sugarcane and its consequences for photosynthesis, biomass partitioning and plant growth. Sci Rep 10, 12492 (2020). https://doi.org/10.1038/s41598-020-69310-6

Viciedo, D.O., de Mello Prado, R., Lizcano Toledo, R. et al. Silicon Supplementation Alleviates Ammonium Toxicity in Sugar Beet (Beta vulgaris L.). J Soil Sci Plant Nutr 19, 413–419 (2019). https://doi.org/10.1007/s42729-019-00043-w

Hurtado, A.C., Chiconato, D.A., Prado, R. de M., Sousa Junior, G. da S., Felisberto, G. Silicon attenuates sodium toxicity by improving nutritional efficiency in sorghum and sunflower plants. Plant Physiology and Biochemistry, 142, 224-233 (2019). https://doi.org/10.1016/j.plaphy.2019.07.010

Teixeira, G.C.M., de Mello Prado, R., Oliveira, K.S. et al. Silicon Increases Leaf Chlorophyll Content and Iron Nutritional Efficiency and Reduces Iron Deficiency in Sorghum Plants. J Soil Sci Plant Nutr 20, 1311–1320 (2020). https://doi.org/10.1007/s42729-020-00214-0

Lines 158 -159. “The omission of N reduced the accumulation of this nutrient in the aerial parts (Fig 1a) and roots (Fig 2a)”. Same issue here, please state those nutrients here.

A: In this sentence we referred to reducing the accumulation of N. We included this in the text and it improved the clarity of the sentence (L 173).

Line 223: Smaller accumulation compared to what?

The presentation of the results in the results section is confusing. The authors indicated percentage reduction in plant biomass for each omitted element, how the percentage reduction was estimated was not clear. What is the reference to which all the treatments were compared to?

Was there any data on the effects on yield? What would be the effects of each deficient nutrient on yield? Since the fruit is most important aspects of this crop, therefore this information would go a long was to really quantify the impact of the study on the local agricultural sector.

A: The rapporteur's suggestions are relevant and important. Estimates of percentage reduction in biomass caused by the omission of each nutrient have always been determined by making comparisons with plants grown in complete solution. We improved the wording of the results section, specifying how each comparison was made (L 212, L 234, L 247, L 262, L 279, L 291, L 295, and L 298).

We understand that production is important information. However, our goal was to determine the effects of nutrient deficiency on this vegetable. In this condition of nutrient scarcity, plants cannot complete the cycle. In classic plant nutrition studies regarding visual diagnosis, most research aims to achieve classic deficiency symptoms (it is the main goal), and associating it with the analysis of nutrients contained in plants. Thus, we set goals similar to these works. Therefore, the yield evaluation evaluated in this manuscript was carried out considering the production of dry mass, which, consequently, will result in production.

Figures

The unit presented are not acceptable in any scientific article. Please consider recalculating these results using a more standardized unit. A possible alternative could be mg/kg of root, shoot etc

A: We understand the reviewer's concern and recognize that we should use standardized units. Thus, we modified the unit of nutrient accumulation data to: “mg per above ground biomass” and “mg per below ground biomass”. To calculate the accumulation of nutrients we multiply the concentration obtained in the aerial part by the dry mass of the aerial part (above ground biomass), and in a similar way for the roots (below ground biomass) (L 138 and L 139). Thus, we chose to use this unit because it is always seen in scientific articles (https://doi.org/10.1371/journal.pone.0250436) and to avoid confusion with concentration data (content) that is usually presented in mg kg-1 or g kg-1 (https://doi.org/10.1371/journal.pone.0123500). The absorption efficiency data are presented in mg g-1 (according https://doi.org/10.1007/s42729-020-00214-0) and efficiency of use in g g-1 (according https://doi.org/10.1371/journal.pone.0240847 and https://doi.org/10.1016/j.plaphy.2019.07.010).

However, if necessary, we are available to make further adjustments to the units.

Reviewer #2: 

General comments: The study is carefully concepted and methodological approach is satisfactory explained. The manuscript deals with interesting and important changes in response to macronutrient deficiency in scarlet eggplant.

Selected parameters are wisely chosen, all of the important nutrients are there, from nitrogen to sulfur. Novelty can be presented from the aspect that such an investigation has never been performed in scarlet eggplant. Results can help to understand the nutritional mechanisms in scarlet eggplant and other plants.

Results and discussion portions are well written. The authors also draw an accurate picture from the results. However, the authors should review and cite some more relevant references. Also needs a proper revision of English language

Many sentences are very confusing. English language and writing style needs to be improved sufficiently

A: We are grateful for the recognition of the scientific merit of our manuscript and for taking the time to read our text. We inform that the text has undergone a rigorous correction, we have included relevant current references, we have improved the clarity of the sentences and the English language has been revised in its entirety. We believe that this revised version is clearer and better than the one previously submitted.

Line: 25- 27. Revise the sentence “but its effects on a plant’s nutritional efficiency to absorb and utilize the missing nutrient, and the reason why these damages reflect in other nutrients have not yet been reported in the culture of scarlet eggplant”

A: The suggestion was answered and the sentence was revised (L 25 to L 27).

Line 46- 47. Do not repeat words of the title in the Keywords

A: Thank you for your suggestion and the authors failed. We include the words "abiotic stress" and "nutritional efficiency" (L 48).

Line: 57-58. Add at least 2 more citation to this line. As many manuscripts have studied plants from different angles, you can cite them here “Such deficiencies can cause a nutritional imbalance, leading to tissues deformities, reduced leaf area, limited growth and dry matter production [4]”.

A: Thank you for your suggestion. We agree that we can reinforce this sentence with more works, so we have included two more relevant and current papers that address the damage caused by nutritional deficiencies in plants (listed below) (L 63).

[7] Osório CRW de S, Teixeira GCM, Barreto RF, Campos CNS, Leal AJF, Teodoro PE, et al. Macronutrient deficiency in snap bean considering physiological, nutritional, and growth aspects. PLoS One. 2020;15: 1–15. doi:10.1371/journal.pone.0234512

[8] Bang TC, Husted S, Laursen KH, Persson DP, Schjoerring JK. The molecular-physiological functions of mineral macronutrients and their consequences for deficiency symptoms in plants. New Phytol. 2021;229: 2446–2469. doi:10.1111/nph.17074

Line: 63-70. As K and N are studied extensively, and many high quality articles are available about this. So, add at least 2 more references here.

A: We agree that we can reinforce this sentence with more work, so we include three more relevant and current papers that address the effects of nutritional deficiencies of N and K on plants (listed below) (L 71 and L 73).

[5] Xie K, Cakmak I, Wang S, Zhang F, Guo S. Synergistic and antagonistic interactions between potassium and magnesium in higher plants. Crop J. 2020;9: 249–256. doi:10.1016/j.cj.2020.10.005

[8] Bang TC, Husted S, Laursen KH, Persson DP, Schjoerring JK. The molecular-physiological functions of mineral macronutrients and their consequences for deficiency symptoms in plants. New Phytol. 2021;229: 2446–2469. doi:10.1111/nph.17074

[10] Ragel P, Raddatz N, Leidi EO, Quintero FJ, Pardo JM. Regulation of K+ nutrition in plants. Front Plant Sci. 2019;10. doi:10.3389/fpls.2019.00281

Line: 108-109. “Its pH was adjusted to 5.5 ± 0.2 using solutions of hydrochloric acid (HCl) or sodium hydroxide (NaOH), both at 1.0 mol/L” replace this sentence by “Its pH was maintained between 5.3-5.7”. Omit the line “using solutions of hydrochloric acid (HCl) or sodium hydroxide (NaOH), both at 1.0 mol/L”.

A: We are grateful for your suggestion and made the change to the text (L 115).

Line: 110-111. Omit the “containing all nutrients”. As you have mentioned complete nutrient solution.

A: We accept the rapporteur's suggestion by making the change to the text (L 116).

Line: 111-113. Mention the particular time (in the form of days) for every nutrient when the symptoms appears.

A: The rapporteur is right and we also believe that it is important to mention the dates of collection of the plants for each nutritional omission. Therefore, we include this information in the text, as follows: 

“After this adaptation period, nutrient omissions were imposed and the plants were then cultivated in this condition until the occurrence of deficiency symptoms that are characteristic of each nutrient. This occurred seven days after the start of omission (DAO) for plants grown in -N (21 days after transplanting - DAT); at 21 DAO for plants grown in -S and -K (35 DAT); at 30 DAO for plants grown in -Ca and -Mg (44 DAT); and at 33 DAO for plants grown in -P (47 DAT).” (L 117 to L 121).

Line: 120-121. All plant materials were washed in tap water, replace with “All plant materials were washed with the tap water

A: Great suggestion and we made the text change (L 129).

Line: 124-125. It’s better to highlight the instrument used for the ionic analysis “The levels of N, P, K, Ca, Mg and S were determined from the digestion of the plant material 125 samples, according to the methods described by Bataglia et al. [10]”

A: We recognize the relevance of the rapporteur's suggestion because we also understand that it is important to provide more details on the method used in determining the nutrient content in plants. Thus, the text was modified as follows: 

“The N content was determined by adding concentrated sulfuric acid to samples, followed by distillation and titration with sulfuric acid [14]. The levels of P, K, Ca, Mg, and S were determined by the digestion of samples, using a digestive mixture of perchloric and nitric acid (1:2), with readings of K, Ca, and Mg performed in spectrophotometry of atomic absorption with air-acetylene flame, while P and S readings were carried out by means of spectrophotometry [14]. The accumulation of each nutrient was calculated based on the plant biomass.” (L 133 to L 139).

Line: 152. Omit the “of Hoagland and Arnon (1950)”, as you have mentioned it in the methodology section. Also follow for all other graphs.

A: Ok. We accept the rapporteur's suggestion and inform that we removed this description from all the captions in the figures.

Line: 161-162. Replaced the sentence “both aerial parts and roots, in comparison to the plants cultivated in the condition of CS” by “both aerial parts and roots, in comparison to the plants under all other conditions.

A: We are grateful for the rapporteur's care in improving the quality of the writing of our manuscript. The text was adjusted according to his suggestion (L 176 to L 177).

Line: 226. Replace “when compared to CS” with “when compared with the CS”

A: We agree with the rapporteur and the text was adjusted according to suggestion (L 240).

Line: 304-405. Please add reference to this statement.

A: Suggestion answered. We have included some references to support our statement (listed below).

[5] Xie K, Cakmak I, Wang S, Zhang F, Guo S. Synergistic and antagonistic interactions between potassium and magnesium in higher plants. Crop J. 2020;9: 249–256. doi:10.1016/j.cj.2020.10.005

[8] Bang TC, Husted S, Laursen KH, Persson DP, Schjoerring JK. The molecular-physiological functions of mineral macronutrients and their consequences for deficiency symptoms in plants. New Phytol. 2021;229: 2446–2469. doi:10.1111/nph.17074

[17] Campos CNS, Teixeira GCM, Prado R de M, Caione G, da Silva Júnior GB, David CHO De, et al. Macronutrient deficiency in cucumber plants: impacts in nutrition, growth and symptoms. J Plant Nutr. 2021;0: 1–18. doi:10.1080/01904167.2021.1921205

Line: 449. If I am not wrong you want to talk about “mRNA”? You mentioned “RNAm”. If this is the case, so revise it please.

A: We appreciate your attention and apologize for the error. In fact, we refer to mRNA, so we made the correction in the text (L 466).

Finally, we emphasize that if further adjustments to the text are necessary to improve its scientific quality, we are available to attend.

Best regards,

The authors

Corresponding author:

Gelza Carliane Marques Teixeira

São Paulo State University (UNESP) School of Agricultural and Veterinarian Sciences, Jaboticabal, Brazil.

Zip code: 14884-900

e-mail: gelzacarliane@hotmail.com

---

## [Decision Letter · Decision Letter 1]

25 May 2021

Nutritional deficiency in scarlet eggplant limits its growth by modifying the absorption and use efficiency of macronutrients

PONE-D-21-00622R1

Dear Dr. Teixeira,

We’re pleased to inform you that your manuscript has been judged scientifically suitable for publication and will be formally accepted for publication once it meets all outstanding technical requirements.

Kind regards,

Umakanta Sarker

Academic Editor

PLOS ONE

Additional Editor Comments (optional): Manuscript may be accepted. However, some typos errors should be addressed well before publication.

Reviewers' comments:

Reviewer's Responses to Questions

**Comments to the Author**

1. If the authors have adequately addressed your comments raised in a previous round of review and you feel that this manuscript is now acceptable for publication, you may indicate that here to bypass the “Comments to the Author” section, enter your conflict of interest statement in the “Confidential to Editor” section, and submit your "Accept" recommendation.

Reviewer #1: All comments have been addressed

Reviewer #2: All comments have been addressed

2. Is the manuscript technically sound, and do the data support the conclusions?

Reviewer #1: Yes

Reviewer #2: Yes

3. Has the statistical analysis been performed appropriately and rigorously? 

Reviewer #1: Yes

Reviewer #2: Yes

4. Have the authors made all data underlying the findings in their manuscript fully available?

Reviewer #1: Yes

Reviewer #2: Yes

5. Is the manuscript presented in an intelligible fashion and written in standard English?

Reviewer #1: Yes

Reviewer #2: Yes

6. Review Comments to the Author

Reviewer #1: Dear Editor,

I am pleased to inform you that the authors have carefully addressed all the questions and comments raised with major improvement compared to the first submission. Specifically the authors have done major gramatical corrections and have improved on the readability and the flow of the manuscript. Therfore, based on the current state of this manuscript, I recommend that the manuscript be accpeted to be published in PLOS ONE journal.

Reviewer #2: All the comments have been justified and the manuscript has been improved significantly. Now it is acceptable for publication. Writing is good but still there are few a typographical mistakes. So, I will suggest authors to thoroughly check the manuscript once.

7. PLOS authors have the option to publish the peer review history of their article (what does this mean?). If published, this will include your full peer review and any attached files.

Reviewer #1: **Yes: **Ibukun Timothy Ayankojo

Reviewer #2: **Yes: **Dr. Sunjeet Kumar

---

## [Editor Report · Acceptance letter]

27 May 2021

PONE-D-21-00622R1 

Nutritional deficiency in scarlet eggplant limits its growth by modifying the absorption and use efficiency of macronutrients 

Dear Dr. Teixeira:

I'm pleased to inform you that your manuscript has been deemed suitable for publication in PLOS ONE. Congratulations! Your manuscript is now with our production department. 

Kind regards, 

on behalf of

Professor Umakanta Sarker 

Academic Editor

PLOS ONE